

# High-resolution sampling and analysis of ambient particulate matter in the Pearl River Delta region of Southern China: source apportionment and health risk implications

Shengzhen Zhou[1,2], Perry K. Davy[3], Minjuan Huang[1], Jingbo Duan[4], Xuemei Wang[1,5*], Qi Fan[1], Ming Chang[5], Yiming Liu[1], Weihua Chen[1], Shanju Xie[6], Travis Ancelet[3], William J. Trompetter[3]

[1] School of Atmospheric Sciences, Sun Yat-Sen University, Guangzhou, 510275, P. R. China

[2] Guangdong Province Key Laboratory for Climate Change and Natural Disaster Studies, Sun Yat-sen University,
Guangzhou 510275, China

[3] National Isotope Centre, Institute of Geological and Nuclear Sciences, 30 Gracefield Road, PO Box 31312 , Lower Hutt, New Zealand

[4] Key Laboratory of Environmental Optics and Technology, Anhui Institute of Optics and Fine Mechanics, Chinese Academy of Sciences, 350 Shu Shan Hu Road, Hefei, Anhui 230031, China

[5] Institute for Environment and Climate Research, Jinan University, Gungzhou, China

[6] Auckland Council, Auckland, New Zealand

Correspondence to: Xuemei Wang (eeswxm@mail.sysu.edu.cn)

**Abstract.** Hazardous air pollutants, such as trace elements in particulate matter (PM), are known or highly suspected to cause detrimental effects on human health. To understand the sources and associated risks of PM to human health, hourly time-integrated major trace elements in size-segregated coarse ($PM_{2.5-10}$) and fine ($PM_{2.5}$) particulate matter were collected at the industrial city of Foshan in the
Pearl River Delta region, China. Receptor modeling of the dataset by positive matrix factorization (PMF) was used to identify six sources contributing to $PM_{2.5}$ and $PM_{10}$ concentrations at the site. Dominant sources included industrial coal combustion, secondary inorganic aerosol, motor vehicles and construction dust along with two intermittent sources (biomass combustion and marine aerosol). The biomass combustion source was found to be a significant contributor to peak $PM_{2.5}$ episodes along with



motor vehicles and industrial coal combustion. Conditional probability function (CPF) was applied to estimate the source locations using the PMF-resolved source contribution coupled with the surface wind direction data. Health exposure risk of hazardous trace elements (Pb, As, Si, Cr, Mn and Ni) and source-specific values were estimated. The total hazard quotient (HQ) of $PM_{2.5}$ was 2.09, higher than the

5 acceptable limit (HQ = 1). The total carcinogenic risk (CR) was $3.37 \times 10^{-3}$ for $PM_{2.5}$, which was three times higher than the most tolerable limit ($1.0 \times 10^{-4}$). Among the selected trace elements, As and Pb posed the highest non-carcinogenic and carcinogenic risks to human health, respectively. In addition, our results showed that industrial coal combustion source was the dominant non-carcinogenic and carcinogenic risks contributor, highlighting the need for stringent control of this source. This study

provides new insight for policy makers to prioritize sources in air quality management and health risk reduction.

## 1 Introduction

Ambient particulate matter (PM) is ubiquitously suspended in the atmosphere, which profoundly

affects human health, visibility and global climate. A number of epidemiologic studies have suggested that short or long-term exposure to PM is associated with a growing risk of the respiratory and cardiac illness, and even premature mortality (Pope III et al., 2009; Dockery et al., 1993). PM mass concentration has been previously considered as the standard metric for protecting human health. More recently, the size and chemical component of PM have been recognized as the important factors for their

toxicity. For instance, Kan et al (2007) found significant associations of daily mortality with $PM_{2.5}$ (fine mode), but not with $PM_{2.5-10}$ (coarse mode) in Shanghai. Reche et al. (2012) also reported higher toxicity for $PM_{2.5-1.0}$ than that of $PM_{2.5-10}$ to human cells. From the epidemiological and experimental findings, it is convincing that trace elements in the PM components appear to be important for causing both pulmonary and cardiovascular disease, such as Ni, V, Pb and Zn (Chen and Lippmann, 2009; Heal

et al., 2012). Moreover, airborne particles and associated trace elements originate from various emission sources, such as motor vehicle, metallurgical industry, coal burning and soil dust, and could emit in a broad size range. Studies suggested that some particle sources are more harmful than others and recommended to control some of the specific sources of PM could be a more effective way for





protecting public health (Bell et al., 2014; Khan et al., 2016). Airborne particle health risk assessment is increasingly depended on the source apportionment of PM using chemical component data. In particular, elemental components are often applied for PM source identification because of their atmospheric stability and source specificity (Taiwo et al., 2014; Visser et al., 2015b).

To identify the sources contributing to measured PM concentrations, multivariate receptor models were used. Positive matrix factorization (PMF) is a powerful and commonly used multivariate receptor model that is capable of resolving factors, or PM sources, without prior source knowledge. It is, however, important to note that source-specific profiles (fingerprints) must be known to properly assign the PMF model outputs. PMF has a number of advantages over traditional factor analysis techniques

including non-negativity constraints and the ability to accommodate missing or below detection limit data. The results of the analysis are directly interpretable as mass contributions to PM from each source factor (Paatero and Tapper, 1994; Paatero, 1997; Song et al., 2001). A majority of PM source apportionment studies in the literature are documented at a lower temporal resolution, typically in 12-24h integrated filter samples (Pant and Harrison, 2012; Belis et al., 2013). Only limited studies have

applied PMF to determine the trace element emission sources at a high time (hourly) resolution (Gao et al., 2016; Dall'Osto et al., 2013; Pancras et al., 2013; Moreno et al., 2011; Crilley et al., 2017; Visser et al., 2015a; Ancelet et al., 2012; Ancelet et al., 2014). Increasing the time resolution of measurements can capture the impact of PM concentration and personal exposure from many intermittent sources such as biomass combustion or industry emission plumes.

The subtropical Pearl River Delta (PRD) region is one of the most urbanized and industrialized areas in China. In recent years, the PRD region is facing severe particulate matter pollution and photochemical smog events (Zhang et al., 2008; Huang et al., 2014b), threatening the health of over 57 million residents (http://www.gdstats.gov.cn/tjnj/2016/directory.html). We have collected hourly samples of fine ($PM_{2.5}$) and coarse ($PM_{2.5-10}$) particulate matter at an important industrial city, Foshan,

in the PRD region. Trace elements were measured by the PIXE technique, providing hourly elemental concentration in the fine and coarse PM. The average concentration, temporal and diurnal variations and effects of meteorology on the trace element concentrations have been reported in Zhou et al.(2016). In this study, we applied a receptor PMF model (EPA PMF 5.0) to characterize the PM sources in Foshan


city utilizing the size-resolved hourly elemental data and other gas phase pollutants. In addition, we assessed the human health risks exposure to the selected trace elements in PM released from the specific sources, as identified by the PMF model. Our results provide valuable information for optimizing the corresponding management and control strategies of PM pollution in the PRD region and cities in other

regions.

## 2. Sampling and analysis

### 2.1 Description of the sampling site

The monitoring station was located in Foshan, Guangdong Province, China (Figure 1), on top of the Foshan Environmental Monitoring Center (Foshan EMC) (Latitude 23.0025º; Longitude 113.1035º,

approximately 35 m above ground level). Foshan is one of the most important manufacturing hubs in China, characterized by the ceramics industry and household appliance industry, and produces ~15% of China's home appliance and ~30% of the world's ceramics (Guo et al., 2011). Adjacent to the monitoring site is a continuous sampling system measuring NOx (Advanced Pollution Instrumentation (API), model 200E), CO (API, Model 300E), SO$_2$ (API, model 100E), O3 (API, Model 300E), and

PM$_{2.5}$ and PM$_{10}$ (Thermo Scientific, model FH62C14), operated by the Foshan Environmental Monitoring Center. The station is surrounded by residential buildings and business offices on flat terrain. Meteorological parameters, including wind speed (WS), wind direction (WD), temperature (T), relative humidity (RH), rainfall were obtained from Foshan Meteorological Bureau. The Meteorological station is situated in the Foshan No. 1 middle school, about 5 km north of Foshan EMC. The time

resolution for the meteorological data is 10 minutes.

### 2.2 Sampling of aerosol

Hourly time-integrated samples of size-segregated coarse (PM$_{2.5-10}$) and fine (PM$_{2.5}$) PM samples were collected using a modified Streaker sampler (PIXE International Corporation, USA ). The Streaker sampler has previously been described in detail (Annegarn et al., 1988), and used in a number of studies

(Annegarn et al., 1992; Annegarn et al., 1996; Filippi et al., 1999; D'Alessandro et al., 2004). The Streaker sampler system used in this study is described in Zhou et al.(2016). Briefly, the Streaker sampler consists of a pre-impactor that removes particles larger than PM$_{10}$ from the incoming air flow, a



thin Kapton foil that collects coarse particles through impaction and a Nucleopore filter (0.4 µm pore size) that collects fine particles. An electronic control system regulates the Streaker sampler pneumatics mass flow (1 litre per minute) and stepper motor rotation of the filter between each user-defined sampling period. The hourly samples were collected with discrete spacing between each deposit to

5 ensure that each deposit was only consisted of particulate matter collected during the intended hour. A total of 60 samples or 60 hours can be collected on each filter.

The high-resolution particulate matter sampling system was installed on the roof of Foshan Environmental Monitoring Centre. The sampler was set to collect hourly samples beginning at 12:00 24/10/2014 and ending at 10:00 14/12/2014. A total of 47 samples or two days were collected per filter

(coarse and fine) with the sampling ending at 11:00 am (47 hours later), allowing an hour to change over filters before the sampling was restarted again at 12:00. In all, 1127 samples each of $PM_{2.5}$ and $PM_{2.5-10}$ were collected representing 1127 hours or seven weeks of sampling.

**2.3 Sample analysis**

Ion beam analysis (IBA) was used to measure the concentrations of elements with atomic numbers

above neon in the PM samples. The IBA was performed using a 3MeV accelerator proton beam with standards ($SrF_2$, NaCl, Cr, Ni, SiO, KCl, Al) run before and after each analytical cycle. Spectral X-ray peak deconvolution was performed using Gupix software (Maxwell et al., 1995). The number of pulses (counts) in each peak for a given element is used by the Gupix software to calculate the concentration of that element. The background and neighbouring elements determine the statistical error and the limit of

detection. Note that Gupix provides a specific statistical error and limit of detection (LOD) for each element in each PM sample. IBA measurements were carried out at the New Zealand National Isotope Centre operated by the Institute of Geological and Nuclear Sciences (GNS) in Gracefield, Lower Hutt, New Zealand (Trompetter et al., 2005). Further details on the IBA techniques used, analytical uncertainties and limits of detection have been reported previously (Ancelet et al., 2012b). Black carbon

was measured using a M43D Digital Smoke Stain Reflectometer (Ancelet et al., 2011).

**2.4 Receptor modeling using PMF**

Receptor modeling and apportionment of PM mass by PMF was performed using the EPAPMF



version 5.0.14 program in accordance with the User's Guide (USEPA, 2015). With PMF, sources are constrained to have non-negative species concentrations, no sample can have a negative source contribution and error estimates for each observed point are used as point-by-point weights. This is a distinct advantage of PMF, since it can accommodate missing or below detection limit data that is a

common feature of environmental monitoring (Song et al., 2001). Prior to the PMF analyses, data and uncertainty matrices were prepared in the same manner as previous studies (Polissar et al., 1998; Song et al., 2001). Data screening and the source apportionment were performed in accordance to the protocols and recommendations set out by Paatero et al, and Brown et al (Paatero et al., 2014; Brown et al., 2015). The effect on the receptor modeling from Variables elements with low signal-to-noise ratios

were excluded from the examined by alternate inclusion and exclusion and only those that could be explained in association with source emissions have been included in the results receptor modeling due to the effect that random analytical noise can have on the receptor modeling process (Paatero and Hopke, 2003).

## 2.5 Conditional probability function (CPF)

In order to locate the local sources and estimate the wind direction impacts on each source identified using PMF, the CPF method was applied. The CPF calculates the probability that a source is located with a particular wind direction sector (Pekney et al., 2006). The probability that a source originates from a given wind direction is estimated by comparing the wind direction distribution for the upper 25 % (or 75 percentile) of source contributions relative to the total wind direction distribution in

this study:

$$CPF_{\Delta\theta} = \frac{m_{\Delta\theta}}{n_{\Delta\theta}}$$

Where:

$m_{\Delta\theta}$ : Number of occurrences from wind sector $\Delta\theta$ for the upper 25 % of concentrations.

$n_{\Delta\theta}$ : Total number of occurrences from the same wind sector.

Using the high temporal resolution data, bivariate polar plots were generated with the R statistical



and Openair software packages (Team, 2011; Carslaw and Ropkins, 2011). Using bivariate polar plots, source concentrations can be shown as a function of both wind speed and direction, providing invaluable information about the direction of potential sources and the influence of wind speed on concentrations. In this study we produced bivariate polar plots using the conditional probabilty function

analysis (CPF), to identify the directions from which high source concentrations are likely to originate. A full description of CPF analysis can be found in Carslaw and Ropkins (2012).

**2.6 Human health risk assessment**

Among the detected elements, As, Pb, Mn, Si and Cr were identified of non-carcinogenic risks; while inorganic As, Pb, Ni and $Cr^{VI}$ were considered carcinogenic (USEPA, 2017).

The adjusted air concentrations ($C_{air-adj}$) for the toxic elements contained in $PM_{2.5}$ was calculated from Equation (1), modified based on the equations in the risk assessment guidance (USEPA, 1989, 2010).

$$C_{air-adj} = \frac{\sum_1^{24}(C_{air} \times ET_i) \times (1day/24hours) \times ED \times EF}{AT \times 1000} \qquad (1)$$

where: $C_{air}$ is the hourly ambient trace elements concentration detected in $PM_{2.5}$ ($\mu g\ m^{-3}$); $ET_i$ is the exposure time (1 hours/day); ED is the exposure duration (6 years for children and 30 years for adults); EF is exposure frequency (350 days/year in this study); AT is the averaging time (30 years×365 days/year for non-carcinogens; 75 years×365 days/year for carcinogens) (Huang et al., 2014a; 2016).

The obtained adjusted air concentrations of the toxic trace elements were subsequently divided by their respiratory reference concentrations ($RfC_s$) to yield the respective hazard quotient (HQ) (USEPA, 2017). Hazard index (HI) was produced by summing up all the hazard quotients (HQs) based on the hypothesis that the adverse effect is proportional to the sum of hazard quotients (HQs). HI less than 1 indicated no significant adverse non-carcinogenic effect.

For the carcinogenic risks assessment, the obtained adjusted air concentrations of the toxic elements of cancer risks from Equation (1) were multiplied by their slope factors to calculate the carcinogenic risks (CRs) (USEPA, 2017). The carcinogenic risk refers to a person's chance of



developing cancer from the exposure to any carcinogenic agent. CR equal to $1 \times 10^{-4}$ is considered as the most tolerable level, above which indicating significant cancer risk.

## 3 Results and discussion

### 3.1 Concentrations of PM$_{2.5}$ and PM$_{10}$ and mass closure analysis

The hourly variations of PM$_{2.5}$ and PM$_{10}$ mass concentrations during the observation are displayed in Figure SM1. The PM$_{2.5}$ and PM$_{10}$ concentrations varied from ~8 to ~270 µg m$^{-3}$ and ~16 to 363 µg m$^{-3}$, with an average value of 62.5 ± 33.5 µg m$^{-3}$ and 91.6 ± 47.2 µg m$^{-3}$, respectively. Around 25% PM$_{2.5}$ mass concentrations exceeded the second grade of new issued National Ambient Air Quality Standard (NAAQS, GB 3095-2012) of China, with daily average value of 75 µg m$^{-3}$. In addition, the mean ratio

of PM$_{2.5}$/PM$_{10}$ was estimated about 0.67, suggesting that fine particles contributed a large part of PM$_{10}$, which is consistent with the previous studies in the Pearl River Delta region, North China Plan and Yangtze River Delta region (Wang et al., 2006; Sun et al., 2004; Hu et al., 2014).

       In order to compare the relationship between the range of analytes measured in PM and the total PM mass a mass closure approach was used. Ideally, when elemental analysis and organic compound

analysis has been undertaken on the same sample one can reconstruct the mass using the following general equation for ambient samples as a first approximation (Cahill et al., 1989; Cohen, 1999; Malm et al., 1994):

$$\text{Reconstructed mass} = [\text{Soil}] + [\text{OC}] + [\text{BC}] + [\text{Smoke}] + [\text{Sulphate}] + [\text{Seasalt}] \quad (3.1)$$

where:

[Soil] = 2.20[Al] + 2.49[Si] + 1.63[Ca] + 2.42[Fe] + 1.94[Ti]

[OC] = Σ[Concentrations of organic compounds]

[BC] = Concentration of black carbon (soot)

[Smoke] = [K] − 0.6[Fe]

[Seasalt] = 2.54[Na]

[Sulphate] = 4.125[S]

       The reconstructed mass (RCM) is based on the fact that the six composite variables or 'pseudo' sources given in equation 3.1 are generally the major contributors to fine and coarse particle mass and



are based on geochemical principles and constraints. As a measure of [OC] was not available in this study, it was assumed that it composed part of the 'remaining mass' (the difference between RCM and gravimetric mass) that includes water and nitrates as major components (Cahill et al., 1989). As Na was below the LOD across most of the samples, the reciprocal calculation of [Seasalt] = 1.65[Cl] was substituted. Most fine sulphate particles are the result of oxidation of $SO_2$ gas to sulphate particles in the atmosphere (Malm et al., 1994). For RCM is assumed that sulphate is present in fully neutralised form as ammonium sulphate. [Sulphate] therefore represents the ammonium sulphate contribution to aerosol mass with the multiplicative factor of 4.125[S] to account for ammonium ion and oxygen mass (i.e. $(NH_4)_2SO_4 = ((14 + 4) \times 2 + 32 + (16 \times 4)/32))$ (Cahill et al., 1989; Cohen, 1999; Malm et al., 1994).

The RCM and mass closure calculations using the pseudo-source and pseudo-element approach are a useful way to examine initial relationships in the data and how the measured mass of species in samples compares to total PM mass, an important consideration for receptor modelling studies. Figure 2 indicates that the RCM of measured elements accounts for approximately 30 % of total mass for both $PM_{2.5}$ and $PM_{10}$ and that there was significant correlation ($r^2$ = 0.64 and 0.69 for $PM_{2.5}$ and $PM_{10}$ respectively), which indicates that the unmeasured PM components (OC, nitrate, bound $H_2O$) are likely to be strongly associated with those species that were measured. Analysis of the data showed that most of the remaining mass (RM) for $PM_{2.5}$ and $PM_{10}$ was associated with $PM_{2.5}$, therefore an extra variable was calculated, $RM_{PM2.5}$ (where $RM_{PM2.5} = PM_{2.5} - RCM_{PM2.5}$) to include in the PMF analysis, an approach that has been successfully applied in other studies (Belis et al., 2013).

**3.2 Sources identification and apportionment of $PM_{2.5}$ and $PM_{10}$**

Using the combined fine and coarse PM elemental datasets along with the gaseous data from the Foshan EMS, PMF was used to identify six factors or sources contributing to PM concentrations during the monitoring period. PMF modeling diagnostics and other parameters are detailed in the *supplementary material.* The sources identified were marine aerosol, biomass combustion, construction dust, motor vehicles and road dust, secondary inorganic aerosol, and industrial coal combustion These sources were found to explain 89% and 91% of the $PM_{2.5}$ and $PM_{10}$ mass, respectively (Figure SM3). The factor profiles are presented in Fig. 3 where elements in the coarse fraction are denoted as *X*-C (e.g. Al-C). The first factor represents biomass combustion because of the presence of BC, K and fine Cl as





primary species along with Zn, which is consistent with previous studies and represents the burning of wood or plant material (Maenhaut, 2017; Ancelet et al., 2012). Motor vehicles and road dust were identified as the source of the second factor based on the presence of $NO_2$, BC and crustal matter components Al, Si, Ca, Fe and Zn from the coarse fraction as the significant elements in the profile.

This profile represents both exhaust (tailpipe) emissions and non-exhaust (road dust and brake and tire wear) emissions, hence the combination of coarse and fine elemental species. Ambient source profiles derived for motor vehicles generally include particulate matter from all engine types as emissions tend to be co-mingled by turbulent air movement at street level due to road traffic and are therefore temporally and spatially covariant (Amato et al., 2009; Pant and Harrison, 2013). The third source

contains most of the black carbon, a substantial amount of CO, $SO_2$, $NO_2$, S and fine fraction heavy metals (Cr, Mn, Cu, Zn, As, Pb) and has been attributed to coal combustion that is probably mixed with industrial process emissions (Song et al., 2007; Tian et al., 2014), The fourth factor contains the majority of both coarse and fine S which dominates the elemental mass and is considered to represent the secondary inorganic aerosol component from gas-to-particle transformation in the atmosphere.

Vanadium and nickel were also associated with this source suggesting contributing source emissions from the combustion of sulfur containing fuels such as heavy fuel oil or oil refinery emissions (Querol et al., 2008; Maenhaut, 2017). Secondary aerosols are primarily fine particles due to the gas-to-particle conversion process but some of the particle size range does extend into the coarse fraction (Anlauf et al., 2006), particularly where heterogeneous atmospheric chemistry takes place on the surface of

particles or in aerosol droplets during the reaction of sulfur gaseous species to form secondary sulfate particle species (George and Abbatt, 2010). The fifth factor, construction dust, with a high Ca loading in the profile along with crustal matter components (Al, Si, Fe) in both the fine and coarse fractions has been attributed to activities that generate cementitious (hence the high Ca content) and crustal matter dusts in the area such as construction/demolition of buildings and other structures (i.e. cement mixing,

concrete pouring, concrete cutting or drilling, and soil excavation) (Owega et al., 2004; Chueinta et al., 2000; Maenhaut, 2017). The sixth source was characterized as marine aerosol due to the predominance of Cl-C. Interestingly most of the $RMPM_{2.5}$ was associated with the motor vehicle and biomass combustion sources (45% and 17%) respectively, most likely indicating the association of OC with



these sources (Querol et al., 2013; Pósfai et al., 2004), and the remainder of RMPM$_{2.5}$ was associated with the secondary aerosol source and is probably indicative of secondary nitrate and secondary organic aerosol concentrations (Huang et al. 2011; Freney et al., 2014).

Average source contributions to particulate mass concentrations are presented in Fig. 4 which shows that industrial coal combustion (32%), secondary inorganic aerosol (28%) and motor vehicle emissions (20%) dominated PM$_{2.5}$ concentrations during the monitoring period. Mass contributions to PM$_{10}$ from coal combustion and secondary inorganic aerosol were essentially unchanged from the fine fraction while the contributions from motor vehicles and construction dust increased due to the significant coarse particle component in those sources.

When considering the gases, SO$_2$ was strongly associated with the industrial coal combustion source, while the majority of NO$_2$ was split between motor vehicles and the industrial coal combustion source. Interestingly CO was primarily associated with the industrial coal combustion and secondary inorganic aerosol sources. The association of CO with secondary inorganic aerosol is explained by co-emission of CO with the gaseous combustion product precursors (e.g. SO$_2$, NO$_x$) of secondary inorganic aerosol and they present in the same air mass. Also included in the source contribution graphs (Fig. 4) is the remaining mass (i.e. measured PM mass – predicted mass by receptor model), The two RM variables RMPM$_{2.5}$ and RMPM$_{10}$, were apportioned to the secondary inorganic aerosol, industrial coal combustion, motor vehicle and biomass burning sources the majority of which was PM$_{2.5}$, and is likely to be associated with the unmeasured components such as nitrate aerosol and primary and secondary organic aerosol. The time-series plots presented in Fig. 5 and Fig. 6 for PM$_{2.5}$ and PM$_{10}$ respectively show that the biomass burning and marine aerosol sources were episodic and that the concentrations of the motor vehicles and construction dust sources increased in PM$_{10}$ due to the coarse particle content.

Peak PM$_{2.5}$ and PM$_{10}$ episodes were associated with motor vehicle and coal combustion particularly when PM from biomass combustion was also a significant contributor. This suggests that biomass combustion sources were additive over the other sources and forced the PM$_{2.5}$ concentrations to extreme highs. Analysis of satellite imagery (MODIS firespot) indicates that biomass combustion activities were outside of the region to the north and northeast of Foshan and were advected over the city during the peak PM pollution events (Figure SM2). The time averaged daily patterns in source



concentrations were examined and as shown in Fig. 7a, the industrial coal combustion source concentrations were generally higher at night, perhaps due to a stable nocturnal boundary layer/mixing height effect. The construction dust concentrations (Fig. 7b) were significantly higher during the day reflecting the pattern of day-time activities generating the dusts. The motor vehicles source (Fig. 7c) showed a bimodal pattern with peaks in the morning and evening characteristic of morning and evening commuter traffic patterns.

The secondary inorganic aerosol source (Fig. 7d) demonstrated no significant observed concentration difference between day and night. This is consistent with the fact that the aerosol formed through secondary processing. Precursor gaseous emissions are oxidized via various gas or aqueous pathways and then converted to sulfate aerosol (Seinfeld and Pandis, 2006; Sun et al, 2014). Therefore, impacts of changes in the boundary layer which would tend to build up other primarily emitted species during the nighttime would be offset by the decrease in the formation rate at night due to the lack of radiation to initiate the oxidation pathways. This is also consistent with the fact that there are a considerable amount of precursor gas emissions from upwind regions and that processing has already occurred there, or while the parcel has undergone dynamical transport. This implies that there is a higher chance of being mixed throughout the boundary layer and lower troposphere as well, and therefore is not as sensitive with respect to a ground-based measurement as more local emissions (Cohen et al., 2011; Cohen and Prinn, 2011). The marine aerosol and biomass combustion sources are not shown in Fig. 7 due to their episodic nature.

Fig. 8 presents bivariate CPF polar plots for the source contributions to PM. Fig. 8a shows that the highest 25% of industrial coal combustion source contributions originated from the west to northwest quadrants under light winds ($1 – 3$ m s$^{-1}$).

A highly resolved spatial anthropogenic emission inventory for PRD was conducted by South China University of Technology for the base year 2012 (Yin et al., 2017). The data shows that the highest density of PM$_{2.5}$ combustion sources (including power plants and industrial combustion) and industrial processes emissions were concentrated within 30 km to the west and northwest of the Foshan EMC as presented in Fig. 9. Three other large coal fired power stations were also located approximately 20 km to the west-southwest of the monitoring site.




Peak marine aerosol (Fig. 8b) arrived at the site from the southeast at higher wind speeds (5 – 7 m s$^{-1}$) in the direction of the South China Sea. Secondary inorganic aerosol (Fig. 8c) came from the north-northeast direction at moderate wind speeds (3 – 5 m s$^{-1}$) with a smaller component from the south-southeast. Construction dusts (Fig. 8d) were highest during low to moderate wind speeds out of the northeast sector, arriving from the direction of the Foshan city centre where the majority of construction activities were occurring. The upper 25% of biomass combustion concentrations (Fig. 8e) were from the northwest sector. It is likely that the biomass combustion emission sources were intermittent and regional, related to agricultural activities or possibly wildfire. Peak motor vehicle source concentrations (Fig. 8f) occurred during south-southwest winds. The PM sampler was located 35m above Fen Jiang Nan Road, which an arterial route with six lanes running north-south past the monitoring site. The CPF result suggests that winds from the south-southwest were most effective in transporting roadway emissions to the sampler.

### 3.3 Health exposure risk of PM$_{2.5}$ elements

### 3.3.1 PM$_{2.5}$ elements human health assessment

Although none of the individual HQ values (As: 0.88, Mn: 0.54, Si: 0.29, Pb: 0.27, Cr: 0.09) for the identified toxic elements exceed 1, the HI value (sum of their HQs) was 2.09, much higher than the safe level (HQ =1). This indicates significant non-carcinogenic risks. Arsenic was observed of the highest risk, followed by Mn, Pb, Si and Cr (Figure 10 (a)). The non-carcinogenic didn't show a distinct diurnal patterns (Figure 11 (a)).

On the other hand, the total CR of the carcinogenic elements was $3.37 \times 10^{-4}$, higher than the most tolerable level of $1.0 \times 10^{-4}$ for adults. Pb ($3.30 \times 10^{-4}$) was found to be the most risky elements (Figure 10(b)). In addition, the CRs of Cr and As were $4.51 \times 10^{-5}$ and $2.27 \times 10^{-5}$, respectively, exceeding the middle tolerable cancer risk level ($1.0 \times 10^{-5}$), and the CR of Ni was $8.71 \times 10^{-7}$, lower than the most acceptable level ($1.0 \times 10^{-6}$). The predominant contribution of Pb to the cancer risks could exactly explain why the diurnal variation of carcinogenic risk was in response to the pattern of Pb ambient air detected concentrations (Figure 11). Around 8:00 AM in the morning was found the most risky time period per day (Figure 11).

### 3.3.2 Health risk assessment of resolved PM sources

Based on the PMF source apportionment results and the contribution of each toxic elements to the non-carcinogenic/carcinogenic risks, the industrial coal combustion sources was identified as the largest non-carcinogenic risk contributor (Figure 12(a)). Besides, construction dust and secondary inorganic

aerosol were also important sources (Figure 12(a)). For the carcinogenic risks, industrial coal combustion was found the largest contributor, followed by biomass burning source. Similar findings were reported by Khan et al (2016) in the a tropical environment in Malaysia. Secondary inorganic aerosols contributed approximately 25% of the $PM_{2.5}$ mass, but they posed little carcinogenic risk. The other three sources (i.e., construction dust, motor vehicles and road dust, and marine aerosol) also

contributed generally low to CR.

From the perspective of human health effects, industrial coal combustion has been identified as the most important emission source. We noticed that it contributed the most to $PM_{2.5}$ mass from the source apportionment results (section 3.2). Therefore, controlling industrial coal combustion sources achieves the benefit of reducing both $PM_{2.5}$ mass concentration and human health risks.

The CPF analysis indicated that the highest health risks of $PM_{2.5}$ came from north and northwest directions (Figure 13). As stated in section 3.2, there are a large number of industrial activities in the north and northwest directions including ceramic industry and coal combustion. Foshan is an industrial city famous in ceramic industry and manufacturing industry (Wan et al., 2011; Tan et al., 2014). The emissions of large quantities of air pollutants from the above-mentioned industries resulted in

deteriorating air quality in Foshan, a city which always ranked among cities with the heaviest air pollution in the PRD region. Since 2007, the Foshan government has launched a series of policy measures to improve the local air quality. The policies are effective in the reduction of PM concentrations, especially in the anthropogenic elements and water soluble ions (Tan et al, 2016). However, our results showed that more efforts are needed to be done to further protect human health.

**4 Conclusion**

In this study, hourly time-resolved major trace elemental composition in coarse ($PM_{2.5-10}$) and fine ($PM_{2.5}$) particulate matter were measured during autumn 2014 at an industrial city Foshan, in the Pearl





River Delta region. $PM_{2.5}$ and $PM_{10}$ mass concentration and some of the gas phase pollutants were concurrently recorded using commercial continuous PM and gases monitors. The results showed that $PM_{2.5}$ and $PM_{10}$ displayed drastic variations with the highest hourly averaged concentration over 250 μg m$^{-3}$ and 350 μg m$^{-3}$, respectively, suggesting the severe PM pollution in autumn/winter seasons at

Foshan city. Source apportionment of PM using positive matrix factorization on the hourly data revealed six $PM_{2.5}$ and $PM_{10}$ sources: industrial coal combustion, secondary inorganic aerosol, motor vehicles and road dust, construction dust, biomass combustion and marine aerosol. Using the hourly resolution elemental data, we improved the source apportionment of PM, especially for the sources with notably temporal variation, such as biomass burning. We found that the industrial coal combustion

sources were the largest contributor to $PM_{2.5}$, while motor vehicles and road dust sources had the highest contribution to $PM_{10}$. Additionally, biomass combustion was observed to additively contribute to $PM_{2.5}$ levels resulting in the highest $PM_{2.5}$ concentrations measured during the monitoring period. Control of regional biomass combustion activities may be one option for preventing extreme $PM_{2.5}$ events and the associated health burden.

Based on the PMF-resolved sources, the health risks posed by selected trace elements (Pb, As, Si, Cr, Mn and Ni) in $PM_{2.5}$ via inhalation exposure were assessed. The results indicated that As and Pb posed the highest non-carcinogenic and carcinogenic risks to human health, respectively. Industrial coal combustion was the dominant source responsible for human health impacts, contributing 66.8% of the hazard index and 92% of the cancer risks. CPF results indicated high health risks in the north and

northwest directions, which was assigned to the intensive industrial activities such as coal burning and manufacturing industry. Therefore, controlling industrial coal combustion sources results in reducing both $PM_{2.5}$ mass concentration and health risk. This study utilized the hourly measured elemental components combined with $PM_{2.5}$, $PM_{10}$ and gas pollutants, and provided valuable information on PM sources identification and control. The source-risk apportionment method helps decision makers to

manage air quality more effectively.





## Acknowledgments

Funding for the program collaboration was provided by the Natural Science Foundation of Guangdong Province (2014A030310497), National Science Fund for Distinguished Young Scholars (41425020), National Natural Science Foundation of China (41505106), Guangdong Provincial
5 Scientific Planning Project (2016B050502005), High-end Foreign Experts Recruitment Program of Guangdong Province (02090-52920003). The New Zealand Ministry of Business, Innovation and Employment, the New Zealand Institute of Geological and Nuclear Sciences, and Foshan Environmental Monitoring center are also acknowledged.




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





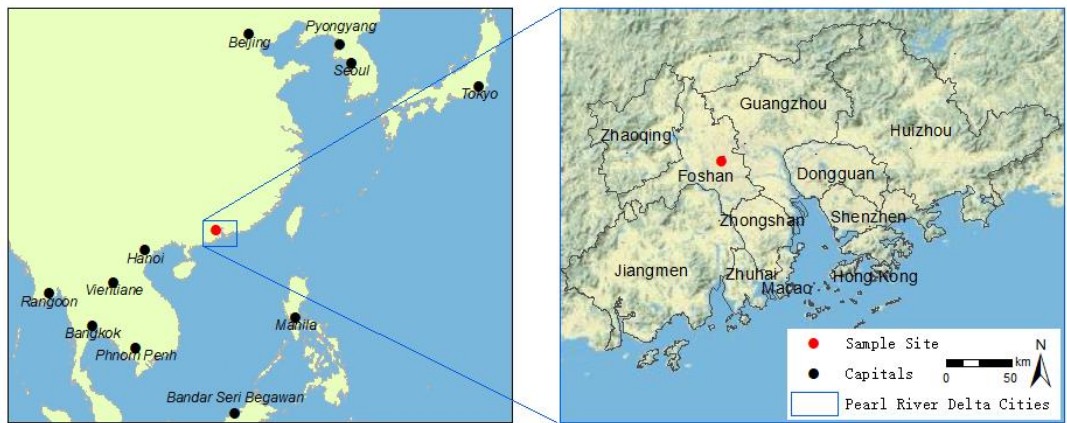

**Figure 1.** Location of sampling site at Foshan City in the PRD Region of Southern China.





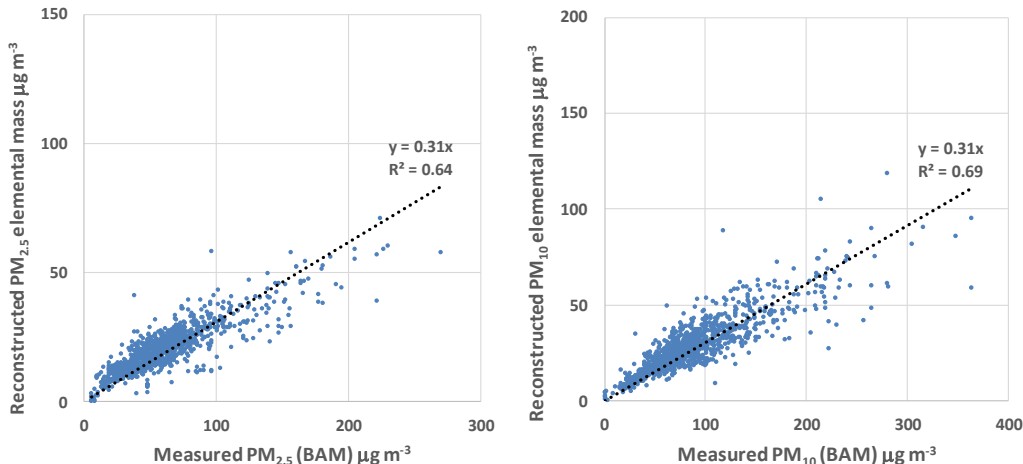

**Figure 2.** Elemental reconstructed mass versus gravimetric mass for (a) $PM_{2.5}$ and (b) $PM_{10}$ at the Foshan site. BAM stands for Beta Attenuation Monitor.





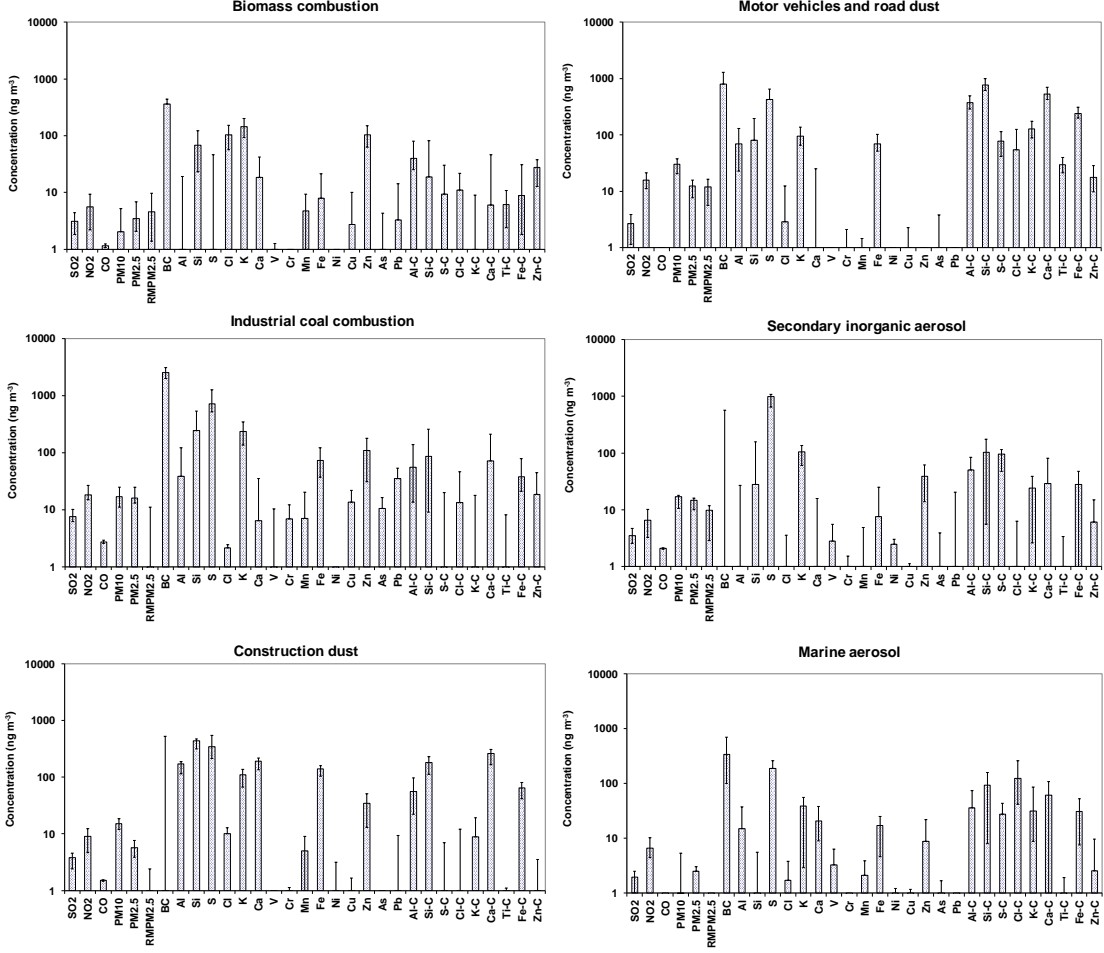

**Figure 3.** Factor profiles for sources at Foshan EMC showing the concentrations of PM$_{10}$, PM$_{2.5}$, SO$_2$, NO$_2$ (all in μg m$^{-3}$) and CO (in mg m$^{-3}$×10). (The error bars are the 5$^{th}$ and 95$^{th}$ percentiles generated from EPAPMF diagnostics).





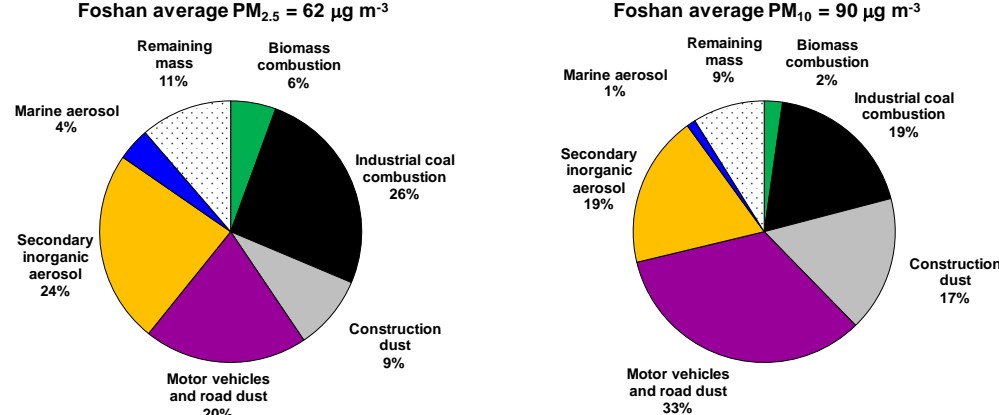

**Figure 4.** Source contributions to PM at Foshan showing the average concentrations of $PM_{2.5}$ and $PM_{10}$ (in µg m$^{-3}$) during the monitoring period.





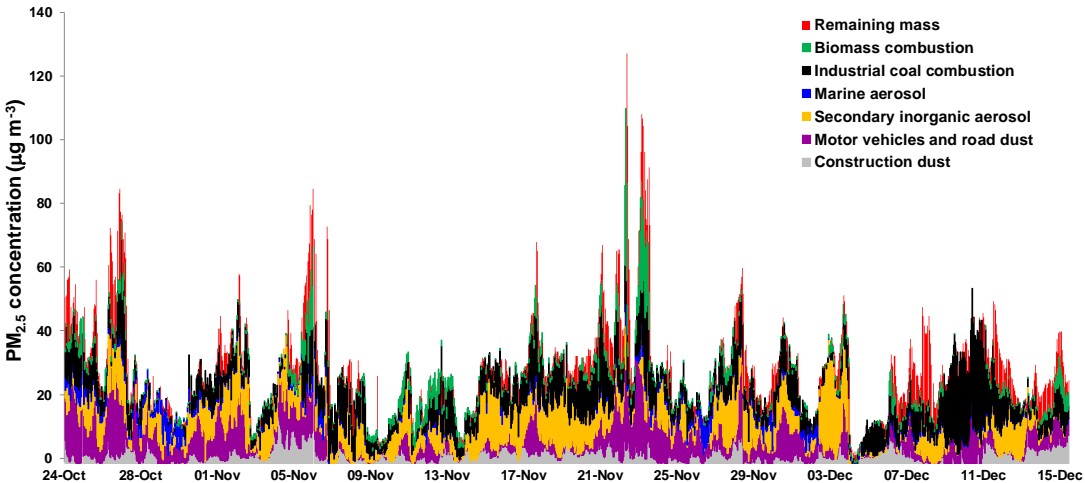

**Figure 5.** Time series for source contributions to PM$_{2.5}$ at Foshan EMC during the monitoring period.





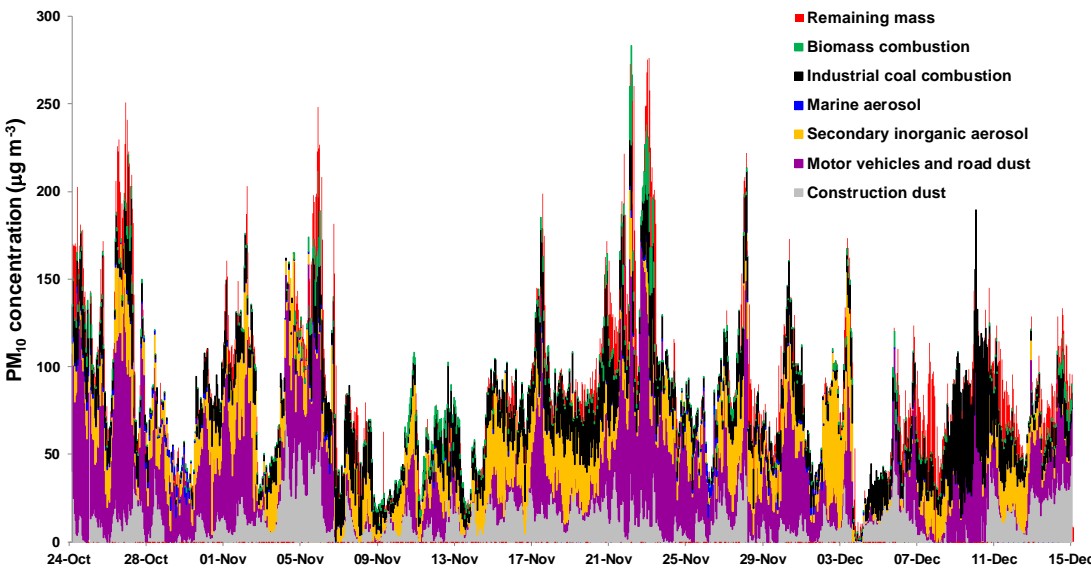

**Figure 6.** Time series for source contributions to $PM_{10}$ at Foshan EMC during the monitoring period.





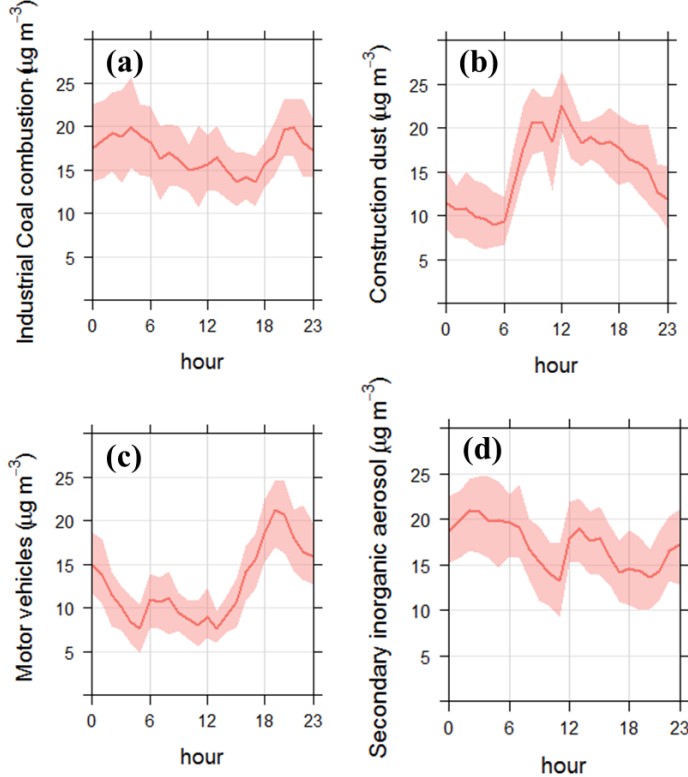

**Figure 7.** Diurnal patterns in source contributions to PM at Foshan EMC during the monitoring period (The shaded area is the 5[th] and 95[th] confidence interval of the calculated mean).





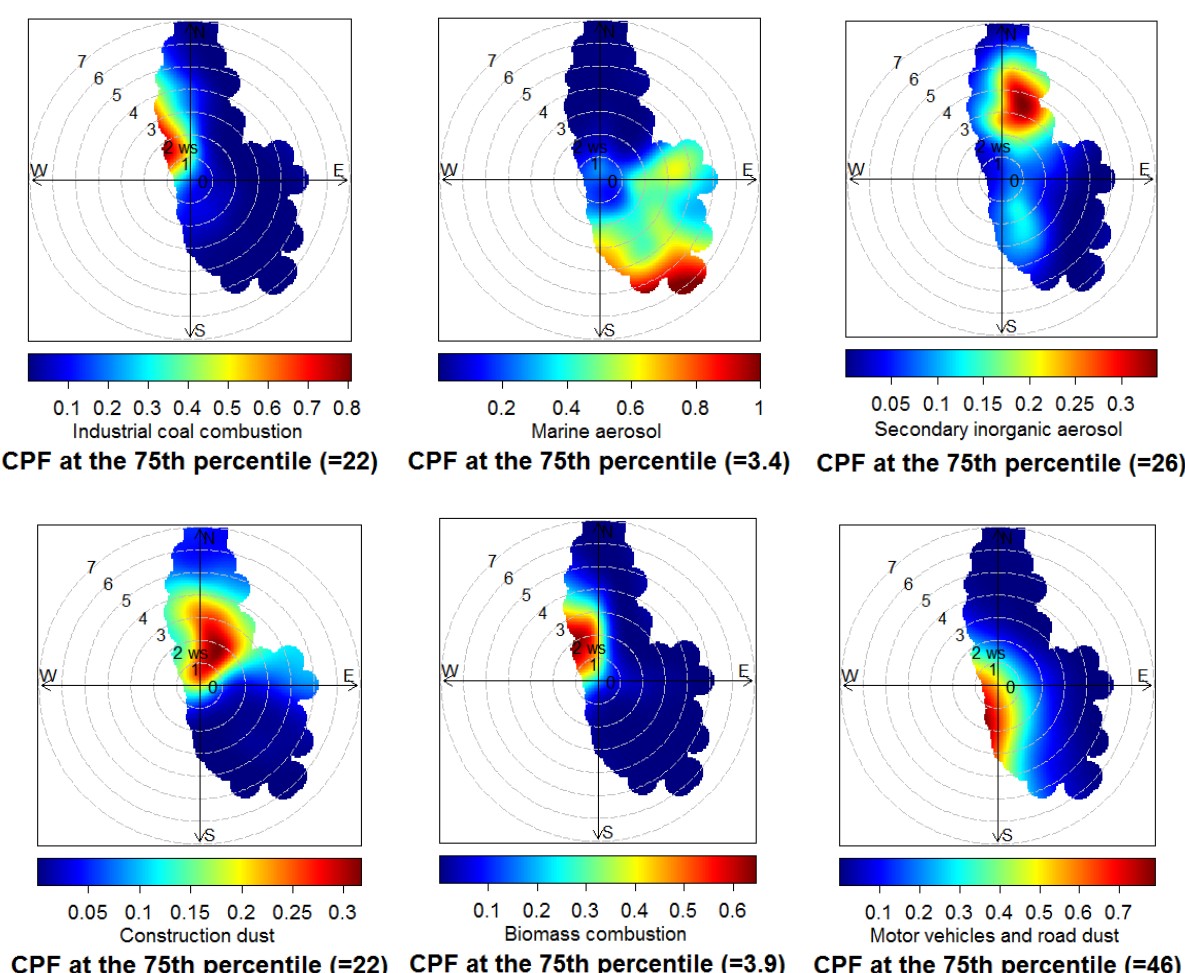

**Figure 8.** CPF bivariate polar plots for source concentrations at Foshan.



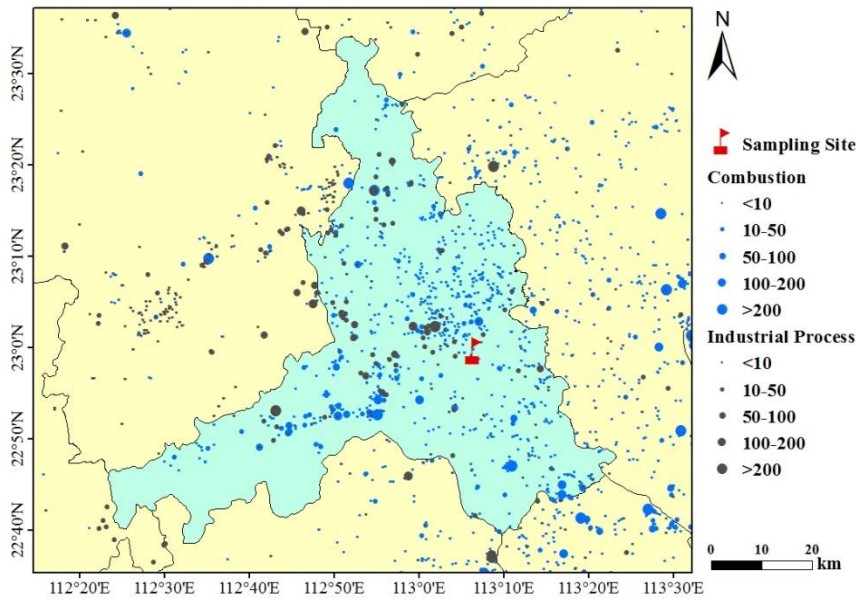

**Figure 9.** Location of PM$_{2.5}$ industrial process and combustion sources in Foshan at the year of 2012 (emissions inventory data in tonnes yr$^{-1}$ from Yin et al.(2017)).



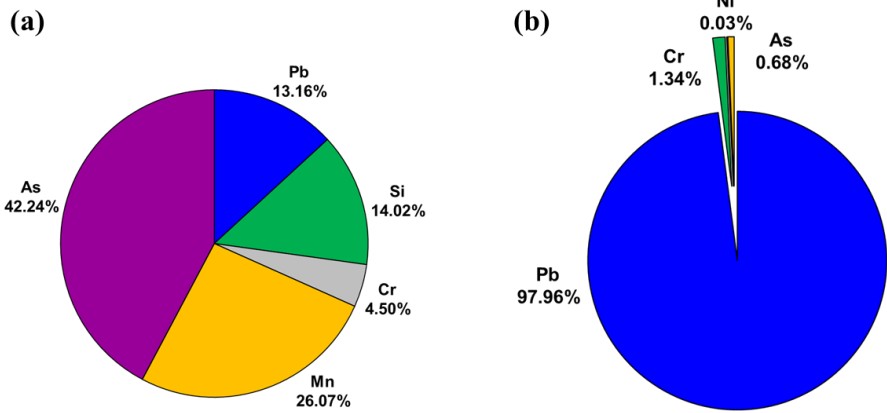

**Figure 10.** (a) average contribution percentage of selected trace elements to Hazard Index (HI) and (b) average contribution percentage of selected trace elements to total carcinogenic risk (CR).



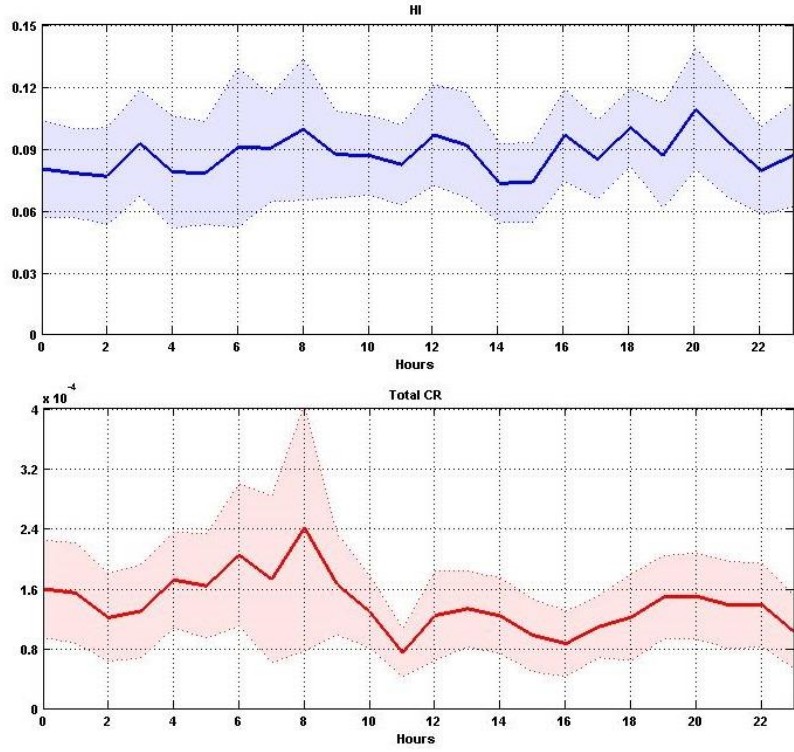

**Figure 11.** Diurnal variations of HI and CR from selected trace elements in PM$_{2.5}$ in Foshan City. Shaded areas
5    represent the 95% confidence intervals.



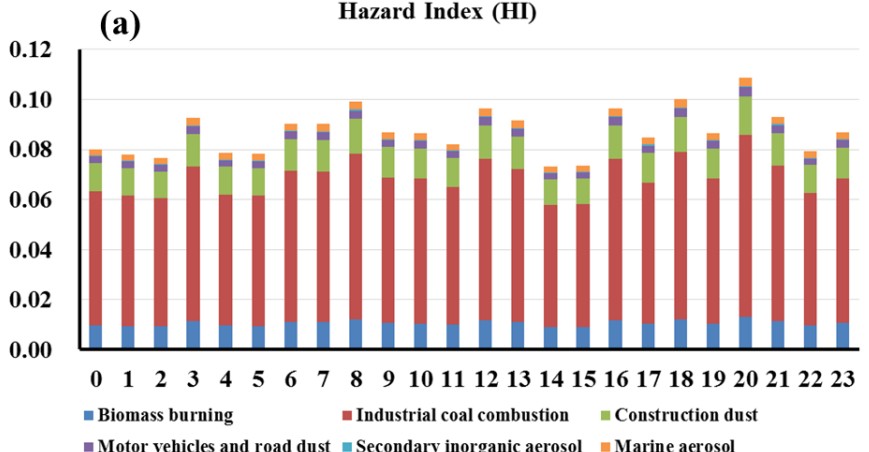

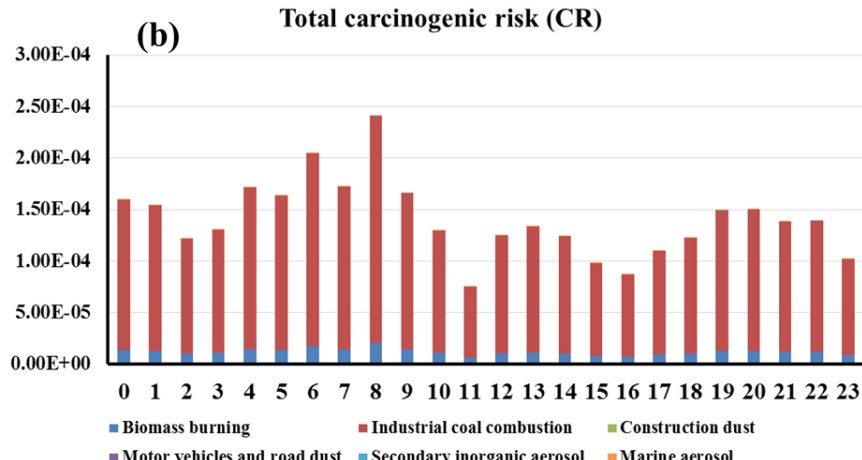

**Figure 12.** Daily variation of average contribution of the identified sources to health risks over the observation periods.




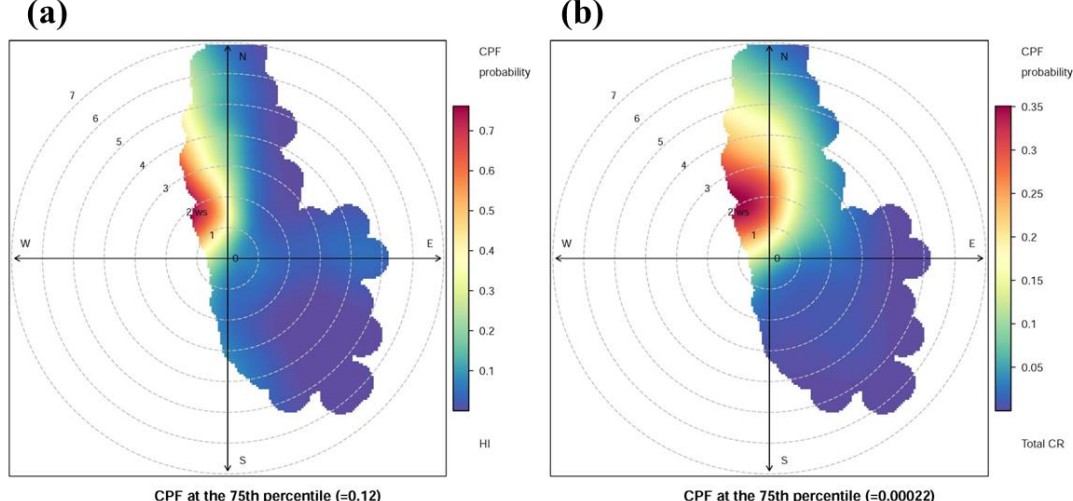

**Figure 13.** CPF analysis polar plots of (a) Hazard Index and (b) total carcinogenic risk at Foshan City. The center of each plot represents a wind speed of zero with increasing radially outward. The HI and CR probability is shown by the color scale.