# Peer review of "High-resolution sampling and analysis of ambient particulate matter in the Pearl River Delta region of Southern China: source apportionment and health risk implications"

_Atmospheric Chemistry and Physics, 2017_

## Referee Comment (RC1) · Anonymous Referee #2 · 7 Nov 2017

General comments

The article submitted is suitable for publication in ACP as the topic is within the scope of the journal. This work presents a detailed interpretation of results obtained from a source apportionment analysis of hourly resolved PM2.5 and PM10 chemical composition, followed by a health risk evaluation of some metals and the obtained sources. The concept is not novel, but high resolution chemical analysis is not performed as a general routine and the important health issues related to coal combustion emissions in an area of high pollution make this work relevant.

[Figure]

The conclusions are clearly drawn from the work, evidencing the importance of pollution and control and emission abatement, especially regarding industrial emissions and coal combustion in the study area.

The title is appropriate, the article is well-structured and the length of the text is adequate. The writing of the article is in general clear (I am not a native English speaker myself). The abstract is clear and concise, the references given throughout the text are appropriate and the supplementary material appropriate.

The scientific methods are clearly explained in the methodology section. However, I have a remark on the use of gaseous compounds together with particulate matter components in the source apportionment analysis. It is not clear to me if the authors use all components only to draw the chemical profiles of the different factors or if they are also used to determine the mass contribution of the sources to the total PM concentrations. This should be clearly explained in the text, and in the latter case, if gaseous compounds and particulate matter components are used together in the analysis this should be discussed carefully when the contributions of the different sources to PM mass concentrations are discussed.

An important drawback of this study is the lack of the measurement of some major compounds, such as organic carbon (OC), nitrate or ammonium. These are related to the non-determined PM mass in the discussion but it is a pity that they are not determined experimentally. This result in important PM components missing from the database for the PMF analysis, which could affect the results obtained. Regarding this issue, there is a paragraph on mass closure with the objective of defining and calculating RM (remaining mass), which is used as an input in the PMF analysis but I would recommend the authors to add more discussion on the limitations related to this.

Taking advantage of the hourly resolution of the measurements, I think it could be interesting to show some results and add some more discussion on the diurnal evolution of hourly resolved components.

As the streaker sampler collects PM2.5-10 and PM2.5 fractions separately, I think it could be interesting to study these two fractions separately, instead of PM10 and PM2.5. Have the authors tried to perform the apportionment analysis with these two fractions?

Specific comments

Page 9, line 5. I suggest the authors to add some discussion on the limitations of indirect calculations. For example sea salt calculated from chlorine could be affected by the volatilization of Cl from the sample.

Page 9, line 17. I suggest the authors to explain better why they use the remaining mass of PM2.5 but not that of PM10 in the PMF analysis.

Page 9, line 21. I think more information should be given on the database used for the PMF analysis. The use of gaseous compounds together with particulate components in the same analysis should be explained more carefully as the source contributions are finally calculated to the total PM mass concentrations.

Page 9, line 26. These sources explain 89% and 91% of the PM2.5 and PM10 mass. . . I don't understand if the gaseous compounds are included in these sources to calculate the total contribution.

Page 11, line 5. These percentages are not the same as in Figure 4. Please correct.

Page 11, lines 6-9. I suggest the authors to give mass concentration values in the text, as it is useful to follow the discussion. I suggest adding them also to Figure 4.

Page 11, lines 13-15. This sentence is not clear. If CO is associated with gaseous combustion products, it should be also associated with the industrial coal combustion and motor vehicle emissions sources. I suggest the authors to explain this.

Page 11, lines 24-16. It is not possible that biomass combustion is also related to domestic heating or industry?

Page 12, lines 4-6. The higher evening peak is also possibly related to a more stable boundary layer. I suggest the authors to explain this.

Page 12, lines 7-8. The concentration of the secondary aerosol source seems to be higher at night.

Page 13, lines 8-9. It is not possible that biomass combustion is also related to domestic heating or industry?

Page 14, lines 4 and 6. It seems that there is some confusion between secondary inorganic and biomass burning source, because they are almost the same color in the graph. Please check and correct.

Figure 2. To my understanding the Beta Attenuation monitor is not a gravimetric method and does not provide gravimetric mass. Please correct.

Figure 3. I suggest the authors to add the %species to the profiles

Figure 4. I suggest the authors to add mass concentrations values in the figure.

Figure 9. This figure could be in the supplementary section.

Figure 12. The colors in the figure are very similar for the biomass burning and secondary inorganic aerosol sources. I think this has produced confusion in the discussion. Please check and correct.

Technical corrections

I have not corrected language issues as I am not a native English speaker myself. However I list here the sentences where I find something that should be checked.

Page 2, lines 6-9. Please check verb tense.

Page 2, line 18. PM mass concentration IS considered as the standard metric for protecting human health.

Page 2, line 22. PM1-2.5 is normally used instead of PM2.5-1

Page 2, lines 27-28. Please check verb tense.

Page 3, line 2. Pease check this sentence: "is increasingly DEPENDED..."

Page 3, line 28. Check word order: "PMF receptor model..."

Page 5, line 5. Check sentence: "......was only consisted"

Page 6, line 8. References are repeated, please correct.

Page 6, lines 9-13. I don't understand this sentence. Please check.

Page 8, lines 9-10. This sentence seems unclear, please check.

Page 11, line 16. Please replace , with .

Page 12, line 20. Figure 8 does not show the letters a, b, c, d. Please correct.

Page 13, lines 1-9. Figure 8 does not show the letters a, b, c, d. Please correct.

Page 13, line 17. "Arsenic was observed of the highest risk". This sentence does not seem correct to me, please check.

---

## Referee Comment (RC2) · Anonymous Referee #3 · 27 Nov 2017

I recommend this manuscript be published after some reversions listed as following. (1) We often believe the coarse PM (PM2.5-10) contributed from vehicles and road dust could probably be crustal dust (almost 18 ug/m3 estimated from Figure 4). Since silicon was measured, please state it is true by using the chemical composition. Also, distinguishing the it from the construction dust is needed. (2) The author insisted coal consumption was dominated by industry, some words are necessary to exclude the power plants in PRD. (3) Some words are needed to tell the readers know how the authors chose the best results from PMF rotation runs.

---

## Referee Comment (RC3) · Anonymous Referee #1 · 27 Nov 2017

The authors conducted a high-resolution measurement of ambient particulate matter in the Pearl River Delta region of Southern China with source apportionment and health risk implications analysis. Overall, the manuscript is well organized and the results are clearly presented with comprehensive interpretation. The analysis of high-time-resolution hazardous elements is limited in published papers. The results will be helpful to understand the source and potential risk of aerosol. The results will also provide important information for policy makers thus such study should be encouraged. I recommend it for publication after addressing the following comments.

[Figure]
* * *
Interactive
comment

Major comments:

It would be helpful if the authors can provide more details on the calculation of uncertainties of elements, and the reasons for choosing the number of factors in this study.

Minor comments:

1. P7 Line 8-9: Cr(VI) and Cr(III) has different health effect but only total Cr is available in current study. Then how was the hazard index for Cr calculated? Similar question for As.

2. P8 Line 17: is it organic carbon (OC) or organic compounds? OM should be used for PM reconstruction.

3. P4 Line 14: O3 should be O3.

4. P9 Line 20: a period is missing.

5. P12 Line 1-2: will the stable nocturnal boundary layer/mixing height affect all sources?

6. P13 Line 24-26: since lead was found to be the most risky elements, the authors may need to provide more information regarding the major source of Pb?

7. P14 Line 1: "the PMF sources. . .", "the" should be deleted.
* * *

---

## Author Comment (AC1) · 1 Jan 2018

**Response to Anonymous Referee #2**

We appreciate your valuable comments and suggestion, which significantly improved the manuscript. We carefully answered them point-by-point as below and improved the corresponding parts in the manuscript.

Reviewer's comments are in plain face.

Author's responses are in blue color.

Changes in the manuscript are in red color.

**General comments**

The article submitted is suitable for publication in ACP as the topic is within the scope of the journal. This work presents a detailed interpretation of results obtained from a source apportionment analysis of hourly resolved $PM_{2.5}$ and $PM_{10}$ chemical composition, followed by a health risk evaluation of some metals and the obtained sources. The concept is not novel, but high resolution chemical analysis is not performed as a general routine and the important health issues related to coal combustion emissions in an area of high pollution make this work relevant.

The conclusions are clearly drawn from the work, evidencing the importance of pollution and control and emission abatement, especially regarding industrial emissions and coal combustion in the study area.

The title is appropriate, the article is well-structured and the length of the text is adequate. The writing of the article is in general clear (I am not a native English speaker myself). The abstract is clear and concise, the references given throughout the text are appropriate and the supplementary material appropriate.

The scientific methods are clearly explained in the methodology section. However, I have a remark on the use of gaseous compounds together with particulate matter components in the source apportionment analysis. It is not clear to me if the authors use all components only to draw the chemical profiles of the different factors or if they are also used to determine the mass contribution of the sources to the total PM concentrations. This should be clearly explained in the text, and in the latter case, if gaseous compounds and particulate matter components are used together in the

analysis this should be discussed carefully when the contributions of the different sources to PM mass concentrations are discussed.

An important drawback of this study is the lack of the measurement of some major compounds, such as organic carbon (OC), nitrate or ammonium. These are related to the non-determined PM mass in the discussion but it is a pity that they are not determined experimentally. This result in important PM components missing from the database for the PMF analysis, which could affect the results obtained. Regarding this issue, there is a paragraph on mass closure with the objective of defining and calculating RM (remaining mass), which is used as an input in the PMF analysis but I would recommend the authors to add more discussion on the limitations related to this.

Taking advantage of the hourly resolution of the measurements, I think it could be interesting to show some results and add some more discussion on the diurnal evolution of hourly resolved components.

As the streaker sampler collects $PM_{2.5-10}$ and $PM_{2.5}$ fractions separately, I think it could be interesting to study these two fractions separately, instead of $PM_{10}$ and $PM_{2.5}$. Have the authors tried to perform the apportionment analysis with these two fractions?

Response: Thank you very much for your comments.

(1) We provide more details on the PMF receptor model, such as the calculation of uncertainties of elements, and the reasons for choosing the number of factors in the section 2.3 and section 3.2. We explained in the text why we used gaseous compounds together with particulate matter components in the source apportionment analysis.

(2) It is better to have additional chemical data, such as organic carbon (OC), nitrate and ammonium to reach a more robust result. We have thought of this problem before and added some discussions in the present version of manuscript (section 3.1, lines 27-).

(3) More information on the diurnal evolution of hourly resolved components can be found in a separate paper (Zhou et al, 2016).

(4). All modelling combinations were run. In fact the fine and coarse elements are

included in this work separately. We have used $PM_{2.5}$ and $PM_{10}$ as the mass variables, as we did not measure coarse PM ($PM_{2.5-10}$) concentrations directly. Unsurprisingly modelling the size fractions separately gives similar results since we do not alter the underlying variance of the data.

Reference:

Zhou Shengzhen, Perry K. Davy, Wang Xuemei*, Jason Blake Cohen, Liang Jiaquan, Huang Minjuan, Fan Qi, Chen Weihua, Chang Ming, Travis Ancelet, William J. Trompetter. High time-resolved elemental components in fine and coarse particles in the Pearl River Delta region of Southern China: Dynamic variations and effects of meteorology. Science of The Total Environment, 2016, 572: 634-648.

**Specific comments**

(1). Page 9, line 5. I suggest the authors to add some discussion on the limitations of indirect calculations. For example sea salt calculated from chlorine could be affected by the volatilization of Cl from the sample.

Response: Thank you for the comments. One paragraph was added to discuss the limitations of this indirect calculation (page 9, lines 13-15).

"though care was taken with the interpretation of this pseudo source due to the potential for Cl loss by volatilization from aerosol (or from filters) in the presence of acidic aerosol species, particularly in the fine fraction (Lee et al. 1999; Chen et al 2016)."

**References:**

Lee, E., C. K. Chan and P. Paatero. Application of positive matrix factorization in source apportionment of particulate pollutants in Hong Kong. Atmos. Environ., 33(19), 3201-3212, 1999.
Chen W.H., Wang X. M., Jason Blake Cohen, Zhou S.Z., Zhang Z.S., Chang M., and Chan C.Y.. Properties of aerosols and formation mechanisms over southern China during the monsoon season. Atmos. Chem. Phys., 2016, 16, 13271-13289.

(2). Page 9, line 17. I suggest the authors to explain better why they use the remaining mass of $PM_{2.5}$ but not that of $PM_{10}$ in the PMF analysis.

Response: We have changed this part and make explanations to this method (page 9, lines 27-): "Analysis of the data showed that most of the remaining mass (RM) for $PM_{10}$ was in fact $PM_{2.5}$, as would be expected for such aerosol species as OC (including secondary organic aerosol) and nitrate. Therefore, an extra variable was calculated, $RMPM_{2.5}$ (where $RMPM_{2.5} = PM_{2.5} - RCMPM_{2.5}$) to include in the PMF analysis as a placeholder for the unmeasured components, an approach that has been successfully applied in other studies (Belis et al., 2013). The advantage of such an approach is that all of PM mass is accounted for in the PMF model. However, the limitation is that it still does not define exactly what aerosol species that the $RMPM_{2.5}$ variable includes, although some inferences can be made depending on the source association."

(3) Page 9, line 21. I think more information should be given on the database used for the PMF analysis. The use of gaseous compounds together with particulate components in the same analysis should be explained more carefully as the source contributions are finally calculated to the total PM mass concentrations.

Response: Thank you for your comments. We provide more details on the calculation of uncertainties of elements, and the reasons for choosing the number of factors in the section 2.3 and section 3.2.

We have used gaseous compounds together with particulate matter components in the source apportionment analysis. This is not a new method, which has been applied in previous studies (e.g., Zhou et al, 2005; Liu et al, 2006). Liu et al (2006) found that inclusion of gas phase data and temperature-resolved fractional carbon data can enhance the resolving power of source apportionment studies. We explain more about this method in the text, and we interpret the results with care as you recommended.

Page 6, lines 6-9: "Another advantage of PMF is that PM mass concentrations can be included in the model as another variable and the results are directly interpretable as the covariant PM mass contributions associated with each factor (source)."

Page 10, lines 15-19: "Note that the gaseous components (SO$_2$, NO$_2$, CO) have been included to aid with source identification and to examine those PM emission sources or secondary aerosol sources they are closely associated (covariant) with. This method has been also adopted by the previous studies, and proved to enhance the resolving power of source apportionment studies (Zhou et al 2005; Liu et al 2006)."

References:

Zhou, L.P.K., Hopke, C.O., Stanier, S.N., Pandis, J.M., Ondov, K., Pancras, J.P. Investigation of the relationship between chemical composition and size distribution of airborne particles by partial least squares and positive matrix factorization. J. Geophys. Res.-Atmos., doi:10.1029/2004JD005050, 2005.

Liu, Wei, Wang, Yuhang, Russell, Armistead, Edgerton, Eric S. Enhanced source identification of southeast aerosols using temperature-resolved carbon fractions and gas phase components. Atmos. Environ, 40: S445-466, doi:10.1016/j.atmosenv.2005.11.079, 2006.

(4) Page 9, line 26. These sources explain 89% and 91% of the PM$_{2.5}$ and PM$_{10}$ mass. . .I don't understand if the gaseous compounds are included in these sources to calculate the total contribution.

Response: We have added some explanation to this method.

Page 6, lines 6-9: "Another advantage of PMF is that PM mass concentrations can be included in the model as another variable and the results are directly interpretable as the covariant PM mass contributions associated with each factor (source)."

Page 10, lines 15-19: "Note that the gaseous components (SO$_2$, NO$_2$, CO) have been included to aid with source identification and to examine those PM emission sources or secondary aerosol sources they are closely associated (covariant) with. This method has been also adopted by the previous studies, and proved to be enhanced the resolving power of source apportionment studies (Zhou et al 2005; Liu et al 2006)."

(5). Page 11, line 5. These percentages are not the same as in Figure 4. Please correct.

Response: Modified in the text.

(6) Page 11, lines 6-9. I suggest the authors to give mass concentration values in the text, as it is useful to follow the discussion. I suggest adding them also to Figure 4.

Response: Thank you for your suggestion. Added in the text and Figure 4.

(7). Page 11, lines 13-15. This sentence is not clear. If CO is associated with gaseous combustion products, it should be also associated with the industrial coal combustion and motor vehicle emissions sources. I suggest the authors to explain this.

Response:

CO, $NO_x$ and $SO_2$ emissions are certainly associated with the **light-duty motor vehicle** source (i.e., petrol vehicles), which emit very few particles directly. The petrol vehicles as well as coal combustion emitted $NO_x$ and $SO_2$ would transform to nitrate and sulfate particles through chemical reactions (secondary inorganic aerosol). Therefore, CO is associated with secondary inorganic aerosol and coal combustion.

**Diesel-powered vehicles** produce very little CO in the first place (Rhys-Tyler, Legassick et al. 2011). It is assumed that diesel vehicle tailpipe emissions are primarily responsible for motor vehicle related $PM_{2.5}$ concentrations, consistent with international research (Kirchstetter, Aguiar et al. 2008, Kim Oanh, Thiansathit et al. 2009, Wang, Tao et al. 2012, Targino, Gibson et al. 2016).

Therefore, CO was primarily associated with the industrial coal combustion and secondary inorganic aerosol sources. Meanwhile, CO is not associated with motor vehicle directly emitted particulate matters (primary aerosol).

We explained more in the text (page 12, lines 13-23):

"CO, $NO_x$ and $SO_2$ emissions are associated with the light-duty motor vehicles (i.e., petrol vehicles), which emit few particles directly. The light-duty vehicles as well as coal combustion emitted $NO_x$ and $SO_2$ would transform to nitrate and sulfate particles through chemical reactions. The association of CO with secondary inorganic aerosol is explained by co-emission of CO with the gaseous combustion product precursors (e.g. $SO_2$, $NO_x$) of secondary inorganic aerosol and they are therefore present (covariant) in the same air mass. Diesel-powered vehicles produce very little

CO in the first place (Rhys-Tyler et al. 2011). It is assumed that diesel vehicle tailpipe emissions are primarily responsible for motor vehicle related $PM_{2.5}$ concentrations in the Foshan urban airshed, consistent with international research (Kirchstetter et al. 2008, Kim et al. 2009, Wang et al. 2012, Targino et al. 2016). Therefore, CO is not associated with motor vehicle emitted particulate matters (primary aerosol)".

References:

Kim Oanh, N. T., W. Thiansathit, T. C. Bond, R. Subramanian, E. Winijkul and I. Paw-armart. Compositional characterization of $PM_{2.5}$ emitted from in-use diesel vehicles. Atmos. Environ. 44(1): 15-22, doi:10.1016/j.atmosenv.2009.10.005, 2009.

Kirchstetter, T. W., J. Aguiar, S. Tonse, D. Fairley and T. Novakov. Black carbon concentrations and diesel vehicle emission factors derived from coefficient of haze measurements in California: 1967-2003. Atmos. Environ., 42(3): 480-491, doi:10.1016/j.atmosenv.2007.09.063, 2008.

Rhys-Tyler, G. A., W. Legassick and M. C. Bell. The significance of vehicle emissions standards for levels of exhaust pollution from light vehicles in an urban area. Atmos. Environ. 45(19): 3286-3293, doi:10.1016/j.atmosenv.2011.03.035, 2011.

Targino, A. C., M. D. Gibson, P. Krecl, M. V. C. Rodrigues, M. M. dos Santos and M. de Paula Corrêa. Hotspots of black carbon and $PM_{2.5}$ in an urban area and relationships to traffic characteristics. Environ. Pollut. 218: 475-486, https://doi.org/10.1016/j.envpol.2016.07.027, 2016.

Wang, R., Tao S., Wang W., Liu J., Shen H., Shen G., Wang B., Liu X., Li W., Huang Y., Zhang Y., Lu Y., Chen H., Chen Y., Wang C., Zhu D., Wang X., Li B., Liu W. and Ma J. Black carbon emissions in China from 1949 to 2050. Environ. Sci. Technol., 46(14): 7595-7603, doi: 10.1021/es3003684, 2012.

(8). Page 11, lines 24-16. It is not possible that biomass combustion is also related to domestic heating or industry?

Response: Satellite fire map showed that lots of fire spots were distributed around the Pearl River Delta region (suburban and rural areas) from November 1 to 31, 2014 over southern China (in supplementary materials).

From our results we can also detect high concentrations of potassium during the fire burning episodes (Zhou et al, 2016).

And also, we can see the obvious burning of crop straw residues alongside the

highway during the autumn season. Therefore, I think it is most likely that the biomass combustion source is associated with agricultural burn-off outside Foshan and would think that very little biomass is used for domestic heating or industry.

We added some explanation in page 13, lines 5-6: "High concentrations of potassium were also detected during the biomass burning episodes (Zhou et al, 2016). It is most likely that the biomass combustion source is associated with agricultural burn-off around Foshan."

(9). Page 12, lines 4-6. The higher evening peak is also possibly related to a more stable boundary layer. I suggest the authors to explain this.

Response: Boundary layer is also an important factor to influence the air pollutant concentrations. We modified this sentence in page 13, lines 10-12: "The motor vehicles and road dust sources (Fig. 7c) showed a bimodal pattern with peaks in the morning and evening rush hours, which ascribed to the morning and evening commuter traffics and lower boundary layer."

(10). Page 12, lines 7-8. The concentration of the secondary aerosol source seems to be higher at night.

Response: Boundary layer is also an important factor to influence the air pollutant concentrations. We modified this sentence in page 13, lines 14-15: "The secondary inorganic aerosol source (Fig. 7d) demonstrated slightly higher concentrations at night than those at daytime."

(11). Page 13, lines 8-9. It is not possible that biomass combustion is also related to domestic heating or industry?

Response: **As explained in the previous question.** Satellite fire map showed that lots of fire spots were distributed around the Pearl River Delta region (suburban and rural areas) from November 1 to 31, 2014 over southern China (in supplementary materials).

From our results we can also detect high concentrations of potassium during the

fire burning episodes (Zhou et al, 2016).

And also, we can see the obvious burning of crop straw residues alongside the highway during the autumn season. Therefore, I think it is most likely that the biomass combustion source is associated with agricultural burn-off outside Foshan and would think that very little biomass is used for domestic heating or industry.

(12). Page 14, lines 4 and 6. It seems that there is some confusion between secondary inorganic and biomass burning source, because they are almost the same color in the graph. Please check and correct.

Response: Thank you for your suggestion. We have corrected in the text.

(13). Figure 2. To my understanding the Beta Attenuation monitor is not a gravimetric method and does not provide gravimetric mass. Please correct.

Response: Thank you and corrected.

(14). Figure 3. I suggest the authors to add the %species to the profiles.

Response: Added in the Figure 3.

(15). Figure 4. I suggest the authors to add mass concentrations values in the figure.

Response: Added in the Figure 4.

(16). Figure 9. This figure could be in the supplementary section.

Response: We have moved this figure in the supplementary section (Figure SM5).

(17). Figure 12. The colors in the figure are very similar for the biomass burning and secondary inorganic aerosol sources. I think this has produced confusion in the discussion. Please check and correct.

Response: Thank you and corrected.

**Technical corrections**

(1). I have not corrected language issues as I am not a native English speaker myself. However I list here the sentences where I find something that should be checked.

Response: Thank you very much for you suggestion. The language has been polished by a native English speaking expert.

(2). Page 2, lines 6-9. Please check verb tense.

Response: Checked and amended.

(3). Page 2, line 18. PM mass concentration IS considered as the standard metric for protecting human health.

Response: Checked and amended

(4). Page 2, line 22. PM1-2.5 is normally used instead of $PM_{2.5-1}$

Response: Checked and amended

(5). Page 2, lines 27-28. Please check verb tense.

Response: Checked and amended.

(6). Page 3, line 2. Pease check this sentence: "is increasingly DEPENDED. . ."

Response: Checked and amended.

(7). Page 3, line 28. Check word order: "PMF receptor model. . ."

Response: Checked and amended.

(8). Page 5, line 5. Check sentence: ". . .. . .was only consisted"

Response: Checked and amended.

(9). Page 6, line 8. References are repeated, please correct.

Response: Checked and amended

(10). Page 6, lines 9-13. I don't understand this sentence. Please check.

Response: Checked and amended in page 6, lines 12-15: "Due to the effect that random analytical noise can have on the receptor modeling process, variables with low signal-to-noise ratios were examined by alternate inclusion and exclusion in a modelling run and only those variables that could be explained in association with source emissions were included in the final results (Paatero and Hopke, 2003)."

(11). Page 8, lines 9-10. This sentence seems unclear, please check.

Response: Checked and amended.

(12). Page 11, line 16. Please replace , with .

Response: Checked and amended.

(13). Page 12, line 20. Figure 8 does not show the letters a, b, c, d. Please correct.

Response: Checked and amended.

(14). Page 13, lines 1-9. Figure 8 does not show the letters a, b, c, d. Please correct.

Response: Checked and amended.

(15). Page 13, line 17. "Arsenic was observed of the highest risk". This sentence does not seem correct to me, please check.

Response: Checked and amended to "Arsenic was observed to have the highest risk……"

---

## Author Comment (AC2) · 1 Jan 2018

**Response to Anonymous Referee #3**

We appreciate your valuable comments and suggestion, which significantly improved the manuscript. We carefully answered them point-by-point as below and improved the corresponding parts in the manuscript.

Reviewer's comments are in plain face.

Author's responses are in blue color.

Changes in the manuscript are in red color.

**Reviewer's comments**

I recommend this manuscript be published after some reversions listed as following.

(1) We often believe the coarse PM ($PM_{2.5-10}$) contributed from vehicles and road dust could probably be crustal dust (almost 18 $\mu g/m^3$ estimated from Figure 4). Since silicon was measured, please state it is true by using the chemical composition. Also, distinguishing the it from the construction dust is needed.

**Response:** We have made some explanations on this problem in the text. One is that the source profiles of the motor vehicle/road dust and construction sources are different. The other is the motor vehicle and construction sources showed distinct time-variation/diurnal concentration patterns from the PMF results (Figure 5, 6, 7). We added more information on this problem.

**Page 10**: "Motor vehicles and road dust were identified as the source of the second factor based on the presence of $NO_2$, BC and crustal matter components Al, Si, Ca, Fe and Zn from the coarse fraction as the significant elements in the profile. This profile represents both exhaust (tailpipe) emissions and non-exhaust (road dust and brake and tire wear) emissions, hence the combination of coarse and fine elemental species and the higher contribution of the source to $PM_{10}$ concentrations relative to $PM_{2.5}$. Ambient source profiles derived for motor vehicles generally include particulate matter from all engine types as emissions tend to be co-mingled by turbulent air movement at street level due to road traffic and are therefore temporally and spatially covariant (Amato et al., 2009; Pant and Harrison, 2013)."

**Page 11**: "The fifth factor, construction dust, with a high Ca loading in the profile along with crustal matter components (Al, Si, Fe) in both the fine and coarse fractions

has been attributed to activities that generate cementitious (hence the high Ca content) and crustal matter dusts in the area such as construction/demolition of buildings and other structures (i.e. cement mixing, concrete pouring, concrete cutting or drilling, and soil excavation), which significantly distinguishes the source from that which might be associated with motor vehicle/road dust emissions (Owega et al., 2004; Chueinta et al., 2000; Maenhaut, 2017)."

**Page 13**: The construction dust concentrations (Fig. 7b) were significantly higher during the day reflecting the pattern of day-time activities generating the dusts. The motor vehicles and road dust source (Fig. 7c) showed a bimodal pattern with peaks in the morning and evening rush hours, which ascribed to the morning and evening commuter traffics and lower boundary layer. We could see the distinguishing diurnal variations between motor vehicles/road dust and construction dust sources.

(2) The author insisted coal consumption was dominated by industry, some words are necessary to exclude the power plants in PRD.

**Response:** Actually, we simply state that this source was likely to be due to coal combustion mixed with industrial emissions so that coal-fired power stations are not specifically excluded - additional wording inserted in **page 11, lines 8-9:**

"The third source contains most of the black carbon, a substantial amount of CO, $SO_2$, $NO_2$, S and fine fraction heavy metals (Cr, Mn, Cu, Zn, As, Pb) and has been attributed to coal combustion which is likely to include coal-fired power station emissions that are probably mixed with industrial process emissions (Song et al., 2007; Tian et al., 2014)."

(3) Some words are needed to tell the readers know how the authors chose the best results from PMF rotation runs.

**Response:** Thank you. New sentences were added in the **section 3.2, lines 11-15:**

"Multiple PMF model runs were performed choosing fewer and more factors to examine the effect on modelling diagnostics and interpretability of the source profiles coupled with the advantage of high-resolution data to examine the diurnal concentration variations. The final six-factor solution adopted included an FPEAK rotation (-3, %dQ(Robust) = 2.91) that provided a good separation of the minor marine aerosol source as evidenced by the FPEAK Bootstrapping results."

---

## Author Comment (AC3) · 1 Jan 2018

**Response to Anonymous Referee #1**

The authors conducted a high-resolution measurement of ambient particulate matter in the Pearl River Delta region of Southern China with source apportionment and health risk implications analysis. Overall, the manuscript is well organized and the results are clearly presented with comprehensive interpretation. The analysis of high time-resolution hazardous elements is limited in published papers. The results will be helpful to understand the source and potential risk of aerosol. The results will also provide important information for policy makers thus such study should be encouraged. I recommend it for publication after addressing the following comments.

Response: We appreciate your useful comments, which improve the quality of the manuscript. We answered them one by one as below.

Reviewer's comments are in plain face.

Author's responses are in blue color.

Changes in the manuscript are in red color.

**Major comments:**

It would be helpful if the authors can provide more details on the calculation of uncertainties of elements, and the reasons for choosing the number of factors in this study.

Response: Thank you for your comments. New paragraphs were added in the section 2.3 (page 5, lines 20-22): "Note that Gupix provides a specific statistical error and limit of detection (LOD) for each element in each PM sample and these have been used to provide the uncertainty matrix used in the PMF analysis."

and section 3.2 (page 10, lines 11-15) to address these problems:

"Multiple PMF model runs were performed choosing fewer and more factors to examine the effect on modelling diagnostics and interpretability of the source profiles coupled with the advantage of high-resolution data to examine the diurnal concentration variations. The final six-factor solution adopted included an FPEAK rotation (-3, %dQ(Robust) = 2.91) that provided a good separation of the minor

marine aerosol source as evidenced by the FPEAK Bootstrapping results".

**Minor comments:**

1. P7 Line 8-9: Cr(VI) and Cr(III) has different health effect but only total Cr is available in current study. Then how was the hazard index for Cr calculated? Similar question for As.

**Response:** Yes, the reference levels and cancer risk slops for Cr vary with its valences (eg., Cr(VI) and Cr(III)) and exposure pathways (eg., oral intake, inhalation, etc.). However, neither reference concentration (Rfc) nor cancer risk slop for inhalation of Cr(III) are available in Integrated Risk Information System (IRIS), so we assumed that the total Cr in our study was Cr(VI), the Rfc and cancer risk slope of which are provided in IRIS (USEPA, 2017).

On the other hand, both the Rfc and cancer risk slop provided in IRIS are specific to inorganic As and its compounds, hence our health risk assessment results for As is specific to the inorganic As compounds, which is the dominating species existing in the airborne particles.

We have added new sentences in the section 2.6, lines 11-16.

2. P8 Line 17: is it organic carbon (OC) or organic compounds? OM should be used for PM reconstruction.
Response: corrected in the text.

3. P4 Line 14: O3 should be $O_3$.
Response: corrected.

4. P9 Line 20: a period is missing.
Response: corrected.

5. P12 Line 1-2: will the stable nocturnal boundary layer/mixing height affect all sources?

Response: Thank you for your comments. Different sources may have their own dynamic variations. Nocturnal boundary layer/mixing height will affect all sources indeed, especially for the industrial coal combustion and motor vehicles sources. We have modified this paragraph.

6. P13 Line 24-26: since lead was found to be the most risky elements, the authors may need to provide more information regarding the major source of Pb?

Response: We added in page 15 (lines 2-3): "From the source apportionment results, lead was mainly emitted from industrial coal combustion (91.5%), and slightly from biomass burning (8.5%)."

7. P14 Line 1: "the PMF sources…", "the" should be deleted.

Response: corrected.